# Evaluating the health and health economic impact of the COVID-19 pandemic on delayed cancer care in Belgium: A Markov model study protocol

**Yasmine Khan**[1,2,3]*, **Nick Verhaeghe**[1,4], **Robby De Pauw**[2,5],
**Brecht Devleesschauwer**[2,6], **Sylvie Gadeyne**[3], **Vanessa Gorasso**[1,2], **Yolande Lievens**[7],
**Niko Speybroek**[8], **Nancy Vandamme**[9], **Miet Vandemaele**[7], **Laura Van den Borre**[2,3],
**Sophie Vandepitte**[1], **Katrien Vanthomme**[1,3], **Freija Verdoodt**[9], **Delphine De Smedt**[1]

1 Department of Public Health and Primary Care, Ghent University, Ghent, Belgium, 2 Department of Epidemiology and Public Health, Sciensano, Brussels, Belgium, 3 Interface Demography, Department of Sociology, Vrije Universiteit Brussel, Brussels, Belgium, 4 Research Institute for Work and Society, KU Leuven, Leuven, Belgium, 5 Department of Rehabilitation Sciences, Ghent University, Ghent, Belgium, 6 Department of Translational Physiology, Infectiology and Public Health, Ghent University, Merelbeke, Belgium, 7 Radiation Oncology Department, Ghent University Hospital and Ghent University, Ghent, Belgium, 8 Research Institute of Health and Society, University of Louvain, Brussels, Belgium, 9 Research Department, Belgian Cancer Registry, Brussels, Belgium

* yasmine.khan@ugent.be

**Data Availability Statement:** No datasets were generated or analysed during the current study. All

## Abstract

### Introduction

Cancer causes a substantial burden to our society, both from a health and an economic perspective. To improve cancer patient outcomes and lower society expenses, early diagnosis and timely treatment are essential. The recent COVID-19 crisis has disrupted the care trajectory of cancer patients, which may affect their prognosis in a potentially negative way. The purpose of this paper is to present a flexible decision-analytic Markov model methodology allowing the evaluation of the impact of delayed cancer care caused by the COVID-19 pandemic in Belgium which can be used by researchers to respond to diverse research questions in a variety of disruptive events, contexts and settings.

### Methods

A decision-analytic Markov model was developed for 4 selected cancer types (i.e. breast, colorectal, lung, and head and neck), comparing the estimated costs and quality-adjusted life year losses between the pre-COVID-19 situation and the COVID-19 pandemic in Belgium. Input parameters were derived from published studies (transition probabilities, utilities and indirect costs) and administrative databases (epidemiological data and direct medical costs). One-way and probabilistic sensitivity analyses are proposed to consider uncertainty in the input parameters and to assess the robustness of the model's results. Scenario analyses are suggested to evaluate methodological and structural assumptions.

relevant data from this study will be made available upon study completion.

**Funding:** - The HELICON project received the grant - The grand number is: B2/202/P3/HELICON - The funder is the Belgian Science Policy Office (BELSPO) within the BRAIN-be 2.0 framework supporting pillar 3 Federal societal challenges - https://www.belspo.be/belspo/index_en.stm - The funders had and will not have a role in study design, data collection and analysis, decision to publish, or preparation of the manuscript.

## Discussion

The results that such decision-analytic Markov model can provide are of interest to decision makers because they help them to effectively allocate resources to improve the health outcomes of cancer patients and to reduce the costs of care for both patients and healthcare systems. Our study provides insights into methodological aspects of conducting a health economic evaluation of cancer care and COVID-19 including insights on cancer type selection, the elaboration of a Markov model, data inputs and analysis.

## Introduction

Every year more than four million incident cancer cases occur across Europe, resulting in around two million cancer deaths [1]. According to the Global Cancer Observatory's 2020 estimations, 13,5 million people were living with cancer over the past five years [2]. Moreover, a total of 47 million disability-adjusted life years (DALYs) were estimated by the Institute for Health Metrics and Evaluation in 2019 for cancer [3]. This has a considerable economic impact. In 2018, the estimated total cost of cancer in Europe reached approximately €199 billion [4]. About €103 billion was spent on direct costs, including €32 billion on cancer drugs, €26 billion on informal care (i.e. the opportunity cost of time forgone by relatives and friends to provide unpaid care) and the total productivity loss was €70 billion [4]. Early diagnosis and timely treatment are key to improving patient outcomes and reducing societal costs [5]. Indeed, delayed cancer diagnosis may lead to a worse prognosis, implying more aggressive and costly treatments affecting patients' quality of life, productivity and survival [6].

Deferred cancer care may be caused by different factors such as a lack of public awareness of the signs and symptoms of cancer affecting care-seeking behaviours [7], outdated and faulty cancer care pathways leading to delayed cancer diagnosis and treatment [8], reduced or unequal accessibility of care [9], poor healthcare coordination [10], and a lack of resources [11]. Unprecedented sanitary situations may also disrupt cancer care as recently seen with the COVID-19 pandemic [6]. Strict COVID-19 measures in hospitals were taken by healthcare providers, supported by professional and scientific societies, which interfered with usual healthcare delivery (i.e. postponement of elective surgeries and deferral of "non-urgent" medical care, change in radiation schedules) [12, 13]. Furthermore, patients feared getting infected by the virus and wanted to avoid overwhelming the healthcare system, which disinclined them from seeking adequate care [14]. Although the health and health economic impact of delayed cancer care during the different COVID-19 pandemic waves is mainly unknown, some studies estimated its partial impact on usual cancer care [15–17]. In Belgium, a 44% decrease in the diagnoses of invasive cancers (i.e. skin melanoma, prostate, colorectal, breast (women), head and neck, haematological malignancies, oesophagus, kidney, bladder, lung, pancreas, cervix) was observed in April 2020 compared with April 2019 [18]. Furthermore, in 2020 the Belgian Cancer Registry (BCR) observed approximately 6% fewer cancer diagnoses than expected compared to 2019 [19]. A recent study conducted by the Madrid tumor registry analysed cancer diagnoses between 2019–2021 in Spain. The study evaluated the differences in annual volume, observed-to-expected (O/E) volume ratios, and standardised incidence rate ratios for 2020–2021 compared to 2019, which was found to be 94.5% (95% CI 93.8–95.3). The study revealed that most cancer types were underdiagnosed in 2020, and this trend worsened in 2021 for colorectal and prostate cancers (87.8%). However, lung cancer showed improvement (102.1%), and breast cancer was overdiagnosed (114.4%) compared to pre-COVID-19

reference data [20]. Another study conducted in Japan found more aggressive and advanced disease after the suspension of breast cancer screening services during the COVID-19 pandemic with the percentage of stage IIB or higher patients being significantly higher in the pandemic group than in the non-pandemic group (22.0% vs 31.3%, p = 0.0133) [21].

Comprehensive quantitative evidence estimating the long-term health and health economic impact of delayed cancer care to guide decision-makers in their decision-making processes is however still scarce. According to a recent British study the COVID-19 pandemic will induce around 33,000 QALY (Quality-adjusted life year) losses and about £104 million in productivity losses in the five years to come [22]. Decision-analytic modelling, such as multistate Markov models (MM), which are typically used to assess the cost-effectiveness of various interventions and programmes in healthcare and public health, can be used as a tool to estimate the costs and outcomes of delayed cancer care [23]. MMs are flexible and therefore ideal to create a methodological framework adaptable to any context and setting. The evidence from such models can guide decision-makers in ways that improve the health outcomes of cancer patients and minimise costs for both patients and healthcare systems.

Hence, this paper aims to provide a MM methodology assessing the impact of delayed cancer care due to the COVID-19 pandemic in Belgium, with a flexible framework that can be used by researchers to answer various research questions in diverging contexts and settings. Our study does not present any result related but merely serves as a methodologic framework.

## Methods

The methods outlined below are based on the Professional Society for Health Economics and Outcomes Research (ISPOR) guidelines [24].

### Study population

For pragmatic and feasibility reasons, a limited number of cancer types are usually selected based on an agreed rationale. This can be carried out by looking at cancer incidence, burden, or prognosis, but in reality such decisions are often based on data availability. In our case, the selection is firstly made by considering the cancers in Belgium with the highest incidence, based on the BCR data [25]. Secondly, by taking into account cancers with the highest DALYs in Belgium [3]. Thirdly, by investigating cancers for which a delay in the initiation of treatment has an important negative impact on the patient's disease progression, metastasis, and survival [6, 26]. To know whether the model is a good reflection of reality, validation is sought by two Belgian experts. Experts were selected using following criteria: at least 10 years of medical practice and active research experience in the field of cancer health economics in Belgium. Expert A has been working as an oncologist for over 20 years and has received health economic training. His research expertise includes the impact of the COVID-19 pandemic on delayed cancer diagnosis and treatment in Belgium. Expert B is an expert in radiation oncology and in health economics. Her work experience includes the organisational aspects of cancer treatment, including the financial and health economic aspects of cancer care. Based on those selection criteria, adult (> 18 years) men and women cancer patients with at least one diagnosis of breast cancer (only women), colorectal cancer, lung cancer, or head and neck cancer are chosen. The methodology outlined below can however be used to assess the impact on other cancer types as well.

### Markov model

MMs are arguably the most popular form of analytical framework used in the health economic evaluation of healthcare interventions. They are generally used to describe random processes

which change over time and to model the course of chronic diseases such as cancer. The disease can be split into different disease states, which are mutually exclusive and exhaustive, implying that each patient depicted can only be in one of these states. Transition probabilities (TP) are used for switching between these states over a discrete time period known as a "cycle" [23]. It is highly recommended that the cycle length is short enough for modifications in subsequent cycles to depict events that change over time [27]. Moreover, a short cycle length reduces errors and simulates events more realistically [23]. MMs do not only describe different health states in a population of interest but can also be used to identify the effects of different policies or therapy options [28]. Those models can predict the long-term costs and outcomes for a modelled cohort of patients (case cohort) by associating estimates of resource use and health outcomes to the model's states and transitions, and then by running multiple cycles [23]. Those costs and health outcomes can then be compared with the experience of a similar (control) cohort, for instance by receiving a different intervention for the same disease (e.g. from the usual care period). In the present study, diagnosed and treated cancer patients from pre-COVID-19 times (control cohort) are compared to diagnosed and treated cancer patients during the COVID-19 period (case cohort). During the pandemic, preventive measures against the virus were put into place which affected the care trajectory of non-COVID patients in a potentially negative way. The purpose of the presented MM is to determine the incremental difference in terms of costs and QALYs, caused by the sanitary crisis. An important characteristic of MMs is that they are without memory, meaning that they are unable to integrate the health experience from previous cycles. However, it is possible to integrate a range of temporary states arranged in a way that each state transitions to the next one. Those states are known as tunnel states because they can only be visited in a predetermined order, much like travelling through a tunnel and an array of tunnel states is used to temporarily change transition probabilities by more than one cycle [27, 29]. If it is impossible to leave a disease state, it is considered to be absorbing (e.g. death) [28].

When developing a MM model, a common first step is consulting published decision-analytic models as a base and source of inspiration. After a thorough literature search of cost-effectiveness studies related to the five selected cancers, a decision-analytic model is developed using a multistate MM approach. Based on the findings from the literature, a draft of a generic multistate MM diagram is made for each considered cancer type (Fig 1). We refer to stages I-IV as defined by the Union for International Cancer Control (UICC), the TNM classification of Malignant Tumours (TNM) (8th edition). The model structure describes cancer progression after diagnosis and subsequent treatment. Therefore a patient diagnosed and then treated with stage I cancer cannot progress to stage II, III, or IV but will evolve in its attributed stage (e.g. if a stage I breast cancer patient is in remission after treatment and after a while the disease progresses, it will not be identified as a stage II breast cancer but as a stage I loco-regional relapse). Patients enter the model at diagnosis. Our MM consists of 18 health states (Fig 1): Missed cancer diagnoses; stage I; stage II; stage III; stage IV; stage I follow-up; stage II follow-up; stage III follow-up; stage IV follow-up; stage I loco-regional relapse; stage II loco-regional relapse; stage III loco-regional relapse; stage IV loco-regional relapse; stage I distant relapse; stage II distant relapse; stage III distant relapse; stage IV distant relapse; dead. It is assumed that once diagnosed, patients are treated using the available and recommended treatment practices. During each cycle, individuals move from stage I, stage II, stage III, or stage IV cancer (i.e. tunnel states), to the corresponding follow-up state (represented by the arrows). During each cycle, patients in the different "follow-up" states can develop a loco-regional relapse or a distant relapse and move to the corresponding states or they can remain in the "follow-up" states. Patients in the different "loco-regional relapse" states can either develop a distant relapse or stay in the same state. According to an expert's opinion, patients being successfully

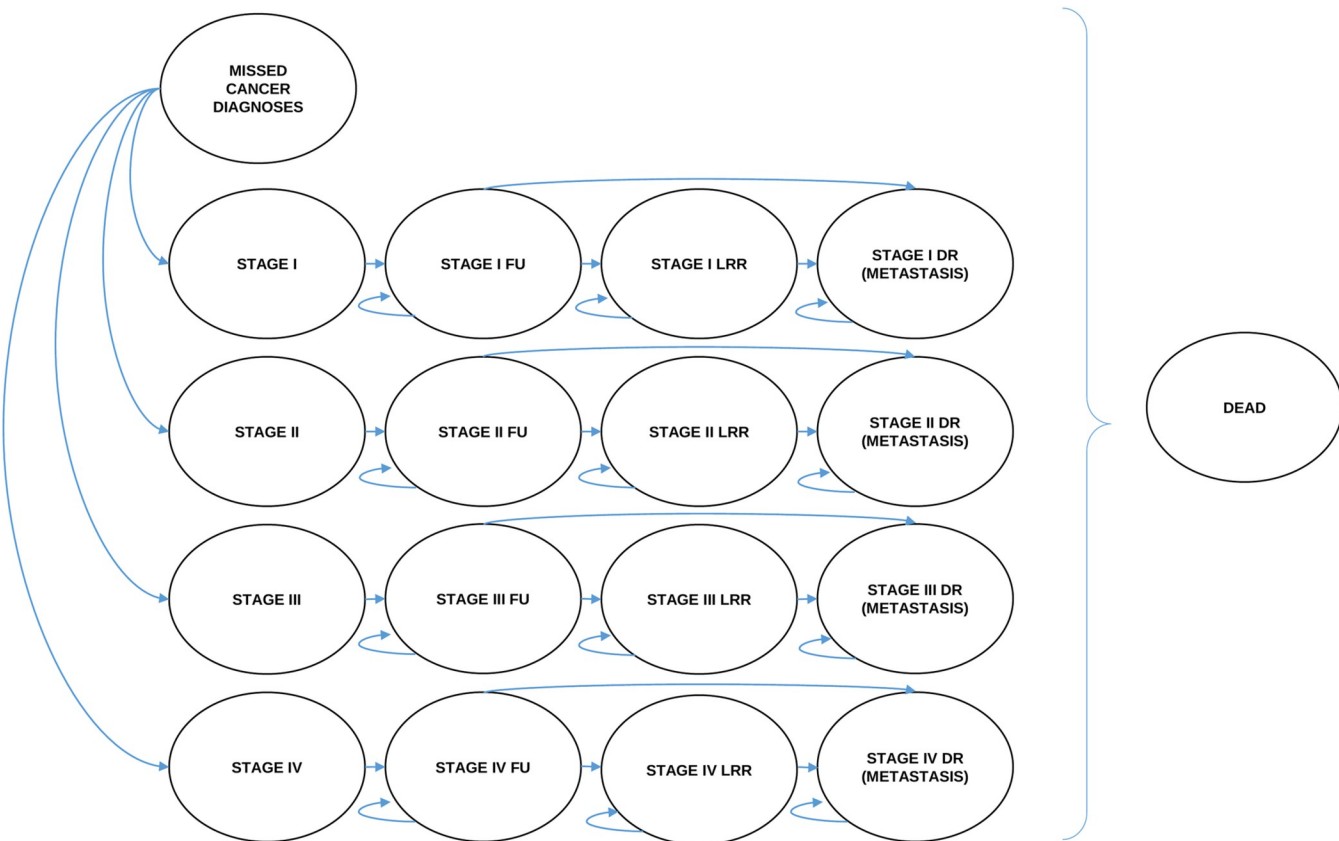

**Fig 1. MM diagram with 18 health states applicable to the five selected cancer types.** FU = Follow-up. LRR = Loco-regional relapse. DR = Distant relapse (metastasis). Curved arrow = Probability of remaining in the state. Straight arrow = Probability of moving to another state. Curly bracket = Probability for patients in any state to move to the dead state.

treated with a loco-regional treatment for loco-regional relapse, will still be considered as in loco-regional relapse state. Patients in the different "distant relapse" states can stay in that state or they can move to the final "dead" state (i.e. absorbing state). Patients being successfully treated with a distant treatment for distant relapse, will still be considered as in distant relapse state. During each cycle, patients in any state can directly move to the dead state. The "missed cancer diagnoses" state represents the cancer patients that are not diagnosed yet due to the disrupted situation and they will either be diagnosed at some time or they will die. The "relapse" state is used as a label to indicate progression (loco-regional and/or distant), and does not mean that patients are not receiving treatment, with patients receiving standard of care treatment regardless of staying in the same state or moving up. Considering the five selected cancers for our study, the natural progression of cancer and the average treatment duration based on published literature, a three-month cycle length is chosen. Researchers may omit and/or adapt steps (i.e. cycle length) described in our paper according to country-specific available data.

## Data inputs

For a MM to be operational several types of data inputs are essential. Model inputs for the decision-analytic model are generally derived from two types of sources: (1) published studies and (2) administrative databases. Cost and epidemiological data should be collected from

country-specific databases, such as, for example, administrative databases and cohort studies. It is also best to use country-specific TPs and utility loss data, however if certain data is not available, it can be derived from international published studies. As recommended by ISPOR-SMDM Medical Decision Making Modelling Task Force, TPs must be collected from studies with comparable settings and populations [30]. In our study, when Belgian data is not available we turn to European countries.

### Transition probabilities

It is unusual for TPs to be extrapolated as such from the literature. Hence, TPs are generally derived from transition rates through two equations allowing the conversion between rates and probabilities [23]. Eqs (a) and (b) show the relation between a probability (p), a constant rate (r), and time (t).

$$r = \frac{-\ln(1 - p)}{t} \tag{a}$$

$$p = 1 - \exp(-rt) \tag{b}$$

In our study, rates are mostly used and are found in studies assessing the risk of cancer relapse and survival and then translated into TPs [31–33].

**Utilities.**  QALYs are calculated by combining utility values based on the EQ-5D index with the time spent in a given health state. The utility values are self-perceived health scores ranging from 0 to 1 with "1" meaning perfect health and "0" dead. One QALY represents one year lived in perfect health [34]. In our case study, utility losses for each cancer state are gathered from systematic reviews of the EQ-5D scores for chronic conditions [35, 36]. Studies using the EQ-5D-5L are preferred over studies using the EQ-5D-3L since they provide increased sensitivity and precision in health status measurement [37]. To account for a particular context and setting, disutility values reflecting disability due to the disease are subtracted from EQ-5D population norm data [35, 38].

**Cost data.**  Costs are calculated based on a societal perspective, which considers both direct and indirect costs. Direct costs represent healthcare expenditures incurred by a cancer patient (i.e. hospitalisation, medical consultations, medication, radiotherapy, therapeutic intervention, surgery) and indirect costs represent costs associated to illness-attributable productivity losses.

In Belgium, cancer data related to the direct costs of each health state are collected through a linked data request destined to the BCR, which collects all new cancer diagnoses with full nationwide coverage, and the intermutualistic agency (IMA), a Belgian organisation which manages the databases containing information on healthcare insurances (compulsory in Belgium) at the population level. In a parallel study, cancer costs are estimated at the individual level for each considered cancer site and each cancer stage with the method of recycled predictions (also known as the method of standardisation or g-computation), which determines the marginal difference in costs for a sample of patients being affected with one type of cancer compared to a sample of the general population who do not present any cancer diagnosis [39–41]. The aim is to estimate the average total healthcare expenditure during one year, being 2018 (the most recent year for which data is available) for cancer patients according to their cancer site and their cancer stage, which will be used as input in the model. The cancer population consists of a sample of the Belgian cancer population alive on the 1st January 2018, and who had at least one cancer diagnosis of the considered cancers during the 10 previous years (S1 Appendix). The control sample is selected from the healthcare insurer database (IMA) if

they were alive on the 1st January 2018 and were not diagnosed with cancer in the 10 years preceding 2018. In order to limit the amount of individuals, a random subset of the patient population is made. Matching is done based on age, sex, and region of residence available on the 1st January 2018. A case-control ratio of 1:4 is used, considering that including more than four controls per case does not increase significantly the statistical power [42].

Because most European cancer registries do not have a direct variable reflecting cancer relapse [43], cost information for the "relapse" states (Fig 1) is collected from the literature, preferably from national studies but when data is not available it is also gathered from international studies. When necessary, costs found in the literature are converted to the national currency and then indexed to the year 2022 by using the Health Index [44]. For feasibility reasons, cost data related to the follow-up of each cancer stage after treatment is also gathered from published studies and is manipulated similarly to cancer relapse costs.

Patients affected with cancer may need time off from the labour force due to treatment (i.e. absenteeism), which represents on the one hand a loss to themselves through a loss of income and on the other hand a loss to society's economy (i.e. productivity). Productivity losses are generally estimated using either the human capital approach (HCA) or the friction cost approach (FCA), which both have advantages and disadvantages [45]. In our study, the FCA is employed because it calculates costs for productivity loss regarding short-term consequences and death, while HCA is more efficient when evaluating lifetime costs [45]. The FCA suggests that society suffers from productivity loss during the time it takes to replace a missing individual (i.e. friction period), with an internal employee in the short term or with a pool of unemployed professionals capable of taking over the job rapidly. It considers the length of the friction period (context-specific and based on unemployment levels) and the ability in identifying replacement workers [45, 46]. Data on productivity loss costs due to cancer are obtained from country-specific published studies. In Belgium, we base ourselves on a Dutch report on methods and costs for economic evaluations in healthcare and assume the friction period to last 160 days [47]. The indirect cost per disease state is estimated by multiplying the number of days off work due to cancer by the average cost for one day of absenteeism [48]. In our study, the number of days off work for each cancer and each disease state is found in published literature. To calculate the cost of death, the average friction period (i.e. 160 days) is multiplied by the mean cost of one day's absenteeism. The costs related to productivity loss are applied to the productive age categories of 30–65 years, considering the proportion of full-time equivalents and the unemployment rate. Costs are then indexed to the year 2022 using the Health Index [44].

**Epidemiological data.** Information on the aggregated number of cancer patients entering the model is derived from national cancer registries. In this study, the aggregated number of stage I-IV cancer patients and the number of missed cancer diagnoses are collected from the BCR.

For our study, a data request has been submitted to the BCR. The number of cancer patients is requested according to variables of interest, in our case being: the cancer site (i.e. breast (C50), colorectal (C18-20), lung (C34), and head and neck (C00-C14; C30-C32)), the timeframe (i.e. pre-COVID and COVID-19 period), the age category (i.e., 18–29; 30–39; 40–49; 50–59, 60–69; 70–79; 80–89; 90+), gender (i.e. male or female for all cancer types except for breast cancer where only females are considered), the region of residence (i.e. Flemish, Brussels-Capital, and Walloon region), and the stage at diagnosis (i.e. stage I-IV and the missed cancer diagnoses). Data from the usual care period (i.e. pre-COVID-19 period) is needed as well as data from the disrupted period (i.e. COVID-19 period), therefore specific timeframes are defined in our data request. The disrupted period is considered from a yearly perspective (i.e. the years 2020, 2021, and 2022) and also according to the different waves and subsequent interwaves of the pandemic (i.e. the first until the fifth wave of the pandemic). The usual care

period is also considered from a yearly perspective (i.e. the year 2019) and according to the pre-COVID-19 dates corresponding to the waves and subsequent interwaves. Seasonal differences in cancer diagnosis are taken into account by comparing the number of cancer diagnosis attributed to each wave and subsequent interwave period (e.g. for the first wave of the pandemic in Belgium: March 1st 2020 until June 22nd 2020) to the number of cancer diagnosis attributed to the corresponding pre-COVID period (e.g. March 1st–June 22nd 2017).

## Analysis

A cohort simulation approach is chosen to analyse the MM. The assumption is that the cohort is distributed between patients within the different initial states of the model (i.e. stage I-IV and the missed diagnoses) at time 0. For every cycle of the MM the correct transition probability is applied and the ratio of cancer patients in each state of the model is adjusted. After numerous iterations of the analysis, a "profile" of the number of patients in each state of the model is developed. As each cancer patient can only be present in one state of the model at any point in time, it is therefore important that for each cycle, the sum of the patients in each health state adds up to the initial number. After multiplying the utility value of the state by the time spent in the actual state and the number of patients from the cohort in that state and adding up the total number of cycles, the model can generate QALY loss estimates. Direct costs are estimated similarly by totalling the costs for patients in each disease state for every cycle, and then by adding up each cycle's costs along all cycles of the model. Two MMs with the same structure are built, one illustrating the usual care period (i.e. pre-COVID-19 times), and the second illustrating the disrupted situation (i.e. COVID-19 period), both having specific input parameter values (Table 1) and a specific starting cohort. The decision-analytic model reports for the selected cancer types (i.e. breast, colorectal, lung, and head and neck) costs (€) and QALY losses and predicts whether or not the disrupted situation compared to the usual care situation engenders increased costs and QALY losses and to what extent. In our case study, the starting cohort of cancer patients is grouped according to their cancer location, cancer diagnosis timeframe, stage at diagnosis, gender, age, and region of residence. For the model to be as realistic as possible and to enable modellers to compare costs and effects in terms of a net present value, it is necessary to make differential time adjustments by discounting costs and effects [29]. Annual discount rates may range from 1.5% to 5% but some countries have different discount rates for costs and effects. In Belgium, health effects are generally discounted at 1.5% and costs at 3.0% [48].

**Time horizon.** The time horizon represents the period over which costs and effects are evaluated [49]. It should be long enough to capture relevant differences in effects and costs among the alternatives examined [50]. Therefore, to evaluate the medium-term health economic impact of the COVID-19 pandemic on delayed cancer care, a 5-year time horizon is preferred. Given the 5-year time horizon and the three-month cycle length, the model runs for 20 cycles and therefore costs and health effects are estimated after 20 cycles.

**Sensitivity & scenario analysis.** A sensitivity analysis is useful to assess how uncertainty in model inputs influences the model outputs. Indeed, input parameter values of MMs can be found in published literature (i.e. clinical trials or observational studies), databases or through expert opinion for example and may therefore be associated with uncertainty. One-way and probabilistic sensitivity analyses (PSA) are performed to consider uncertainty in the input parameters and to assess the robustness of the model's results. A one-way sensitivity analysis is a deterministic sensitivity analysis where one parameter is varied at a time to see how the model results are affected [51]. For each cancer type, the aggregated number of cancer patients entering the model for each stage at diagnosis, transition probabilities, utilities, direct costs,

**Table 1. Input parameters for which a value is needed for the MM to be operational.** Each cancer (i.e. breast, colorectal, lung, and head and neck) has its own values.

| Aggregated number of initial cancer patients | Value | Transition probabilities | Value | Utilities | Value | Direct costs (hospitalisation, appointments, drugs, radiotherapy, surgery, therapeutic intervention) | Value | Indirect costs (Productivity loss) | Value | Discount rate | Value |
|---|---|---|---|---|---|---|---|---|---|---|---|
| Missed cancer diagnoses | | Stage I to Stage I FU | | Stage I | | Stage I | | Cost per day absenteeism | | Costs | |
| Stage I | | Stage I to Stage I LRR | | Stage I FU | | Stage I FU | | Number of days off work for stage I cancer | | Effects | |
| Stage II | | Stage I to Stage I DR | | Stage I LRR | | Stage I LRR | | Number of days off work for stage II cancer | | | |
| Stage III | | Stage I to Dead | | Stage I DR | | Stage I DR | | Number of days off work for stage III cancer | | | |
| Stage IV | | Stage II to Stage II FU | | Stage II | | Stage II | | Number of days off work for stage IV cancer | | | |
| | | Stage II to Stage II LRR | | Stage II FU | | Stage II FU | | Death (based on the Friction Cost Approach) in days | | | |
| | | Stage II to Stage II DR | | Stage II LRR | | Stage II LRR | | | | | |
| | | Stage II to Dead | | Stage II DR | | Stage II DR | | | | | |
| | | Stage III to Stage III FU | | Stage III | | Stage III | | | | | |
| | | Stage III to Stage III LRR | | Stage III FU | | Stage III FU | | | | | |
| | | Stage III to Stage III DR | | Stage III LRR | | Stage III LRR | | | | | |
| | | Stage III to Dead | | Stage III DR | | Stage III DR | | | | | |
| | | Stage IV to Stage IV FU | | Stage IV | | Stage IV | | | | | |
| | | Stage IV to Stage IV LRR | | Stage IV FU | | Stage IV FU | | | | | |
| | | Stage IV to Stage IV DR | | Stage IV LRR | | Stage IV LRR | | | | | |
| | | Stage IV to Dead | | Stage IV DR | | Stage IV DR | | | | | |
| | | Missed cancer diagnoses to Stage I | | | | Missed cancer diagnoses | | | | | |
| | | Missed cancer diagnoses to Stage II | | | | | | | | | |
| | | Missed cancer diagnoses to Stage III | | | | | | | | | |
| | | Missed cancer diagnoses to Stage IV | | | | | | | | | |

FU: Follow-up

LRR: Loco-regional relapse

DR: Distant relapse

indirect costs, and discount rates are included in the sensitivity analysis. A tornado diagram is then used to indicate which of the one-way sensitivity analyses influences the model results the most, with each bar representing a one-way sensitivity analysis and their width illustrating the impact on model results [51]. Since one-way sensitivity analyses underestimate uncertainty because they assume that uncertainty exists only in one parameter, it is useful to also perform a PSA [51]. In a PSA, random sampling from each distribution is used to create a set of input parameter values, the model is then "run" to generate outputs, which are stored. The procedure may be repeated several hundreds or thousands of times, which results in a distribution of outputs that can be visualised on a cost-effective plane, and assessed [52]. As a general recommendation, beta distributions are used for proportions and ratios, gamma for right-skewed parameters (i.e. relative risks, hazard ratios, and odds ratios) [51]. Therefore, in our case study, gamma distributions are used for the aggregated number of cancer patients entering the model for each stage at diagnosis and the costs, beta distributions for transition probabilities, and utilities (which can take a negative value).

Scenario analyses are performed to evaluate methodological and structural assumptions. In our case, at least three additional scenarios will be tested. The first scenario uses a different time horizon of 15 years. This is because for some major cancers such as breast cancer, the cumulative incidence of late breast cancer relapse is 8.5% (95% CI = 8.1% to 8.9%) 15 years after primary diagnosis [53]. It is therefore worth estimating the impact of delayed cancer care on late relapses. The second scenario uses an age-specific model for cancer patients aged between 45 and 64 years, this is because 34% of the estimated new cancer diagnoses and 25% of estimated deaths occur in this age group in Europe [54]. Furthermore, since those patients are part of the workforce it is relevant to assess productivity losses associated with their absence due to the disease. The final scenario consists in using a 0% discount rate for both costs and benefits, which will allow decision makers to judge the importance of using different discount rates for the final result [48].

Our study was not submitted to nor approved by any institutional ethics committee because our study does not involve human participants, human data or human tissue.

## Discussion

Considering that the health and economic burden of cancer is more than substantial [4], it seems crucial to prioritise resource allocation considering the needs of patients, society, and healthcare systems based on health economic evidence. Given its adaptable and accessible features, decision-analytical modelling, and particularly multistate MM, can be used to unravel the health and health economic impact of delayed cancer care in diverse contexts and settings. Through our study focusing on the COVID-19 pandemic and Belgium we offer a MM methodology assessing the impact of delayed cancer care that can be applied to address a variety of research questions in a range of settings and situations. Decision-makers may be particularly interested in the findings of such health economic analyses since they will help them decide how to deploy resources to enhance cancer patients' health outcomes and keep costs of care down for both patients and healthcare systems.

In 2016, the World Health Organisation predicted an 18 million shortage of healthcare workers by 2030 [55]. The growing demand for cancer treatment in the past decades when treatment capacity had not increased proportionally enough may partly explain delayed cancer care and therefore play a substantial part in the economic burden of cancer [11, 56]. The recent COVID-19 pandemic caused further disruptions in the cancer care sector which aggravated the baseline situation [16, 17, 57]. The National Institute for Public Health and the Environment (RIVM) in the Netherlands published multiple reports where the indirect impact of the

pandemic on delayed care of various diseases are estimated. The RIVM were asked to model in one of their reports, the expected effect of delayed care for skin melanoma on quality of life based on a shift in the stage at which the cancer is diagnosed and treated [58]. Two scenarios were tested with three and six months delayed diagnosis and treatment for a time horizon of five and ten years. Results showed that in a three-month delay scenario patients diagnosed with a certain stage would progress to the next stage (i.e. stage I to stage II). In the latter scenario, the stage shift would result in 549 and 1,229 years of life lost and 853 and 1,704 QALY losses respectively over a five and ten-year time horizon. However, for the six-month delay scenario, patients would advance two stages (i.e. stage I to stage III) which would result in 1,058 and 2,432 years of life lost and 1,454 and 3,021 QALY losses respectively over a five and ten-year time horizon. Those estimations appear substantial, however, it may take a few years before being able to quantify the actual extent of the collateral damage caused by the sanitary crisis for cancers.

A recent systematic review and meta-analysis evaluating the impact of delay in treatment on the risk of death across seven types of major tumours (i.e. bladder, breast, colon, rectum, lung, cervix, and head and neck) found that a four-week treatment delay was associated with a 6 to 13% increased risk of death, depending on the treatment modality [6]. Elective surgery was halted for more than four weeks in most countries during the first wave of the COVID-19 pandemic, delaying the treatment of cancer patients [6]. It therefore seems important to evaluate the collateral damage that the sanitary crisis may have caused on cancer care.

Researchers conducting their study in Belgium or similar settings may find the present study useful in terms of administrative data collection. Indeed, it may allow one to know which data holders to consult when conducting a health economic evaluation on cancer or which variables to select from a particular database when initiating a data request.

The present study may also provide scientists conducting a health economic evaluation in a context including cancer and COVID-19 with insights on cancer type selection, the elaboration of a MM, data inputs, and analysis.

There are multiple applications possible for the type of model exposed in this study other than the COVID-19 pandemic and Belgium. One could investigate the impact of errors in cancer diagnosis on delayed cancer care. The disrupted situation would be compared to the usual care situation (i.e. no error in diagnosis) and the potential increased costs and negative impact on the quality of life of the patient due to mistakes in diagnosis could guide decision-makers in allocating resources in technologies that have proven to decrease those lapses. Another interesting application of this model could be to evaluate the impact of an innovative cancer screening test compared to the standard test or no cancer screening on delayed cancer care in a specific country, for cancers that are typically not diagnosed until they are at an advanced stage. The estimated costs and health effects of the intervention could allow decision-makers to either opt for a new screening test, the standard test or no screening, based on the most cost-effective option in a specific context and setting. Health economists using the present MM protocol may attribute value to their final model by finding comparable results in similar published studies.

We acknowledge some limitations within our study. First of all, non-invasive cancer (stage 0), which may progress into a further cancer stage, is not taken into account, when in reality, a proportion of the population is affected by it. The reason for this is the lack of published evidence related to MM input parameters such as TPs, utilities, and costs. Hence, depending on the proportion of the population presenting non-invasive cancer costs and QALY losses may be underestimated. Another important limitation is that the difference in cancer cases observed when comparing the usual care period to the disrupted situation may not entirely be due to the disrupted situation. Indeed, even during usual care periods there may be several

potential delays in the cancer diagnostic and/ or treatment pathway (i.e. patient delay, primary care delay and secondary care delay), which may also explain the potential decrease in cancer cases [8]. A further limitation is that the present study aims to provide a MM methodology that can be adapted to different studies in different countries, however as already stated by Willems et al., accessibility to health economic data differs by country [59]. Although clinical data may be extrapolated across countries, economic information depends on a country's characteristics, therefore data imputation may occur. The availability and timeliness of a country's epidemiological and cost data represent another limitation. Even in developed countries, centralised data is not always accessible, necessitating the completion of strenuous and time-consuming administrative tasks (i.e. data linkages along with data requests). The costs and barriers related to reusing these administrative and health data, privacy concerns, and in the case of individual linkage, the administrative burden must not be forgotten.

MMs have some inherent limitations, such as the fact that they always depict a simplification of reality. Moreover, there is uncertainty linked to model parameters (although they are taken into account in the PSA) and modelling assumptions, such as the time horizon and the cycle length [23]. In our study, we account for parameter uncertainty as well as scenario uncertainty, trying to summarise at best the uncertainty coming from methodological choices. An essential and recognised limitation of a MM is its "memoryless" feature also called the "Markovian assumption", being that the probability of moving from one health state to another is not influenced by the patient's history [23]. However, this constraint may be overcome by using time-dependent TPs and by using a combination of separate states to model specific patient histories. An additional element to be cautious about is the calculation of TPs for the MM, indeed probabilities found in the literature may not refer to the same cycle length selected for the MM. It is therefore necessary to use adequate equations [23]. Additional elements of a MM may be open to criticism such as its difficulty to simulate interactions between individuals and the environment as well as its lack of ability to manage multiple events simultaneously [23]. Such limitations can be overcome by using agent-based models.

Since cancer already carries a significant health and economic burden, it appears essential to stop the problem from getting worse by, for example, addressing delayed cancer care. Using health economic evaluations in decision-making could result in allocating resources in ways which improve health outcomes of cancer patients and minimise costs for patients, society and healthcare systems. Multistate MMs are analytical frameworks well used in decision analysis, adaptable to various contexts and settings and are well used to describe chronic conditions such as cancer. A future paper is aimed to be published using the presented methodology and as yet unavailable Belgian data, which will allow for estimating the medium-term health economic impact of the COVID-19 pandemic on delayed cancer care in Belgium.

## Supporting information

**S1 Appendix. Cancer patient selection for the estimation of the direct costs of cancer.** (DOCX)

## Acknowledgments

We would like to thank the cancer experts who have kindly accepted to review our study protocol.

## Author Contributions

**Conceptualization:** Yasmine Khan, Nick Verhaeghe, Delphine De Smedt.

**Formal analysis:** Yasmine Khan.

**Investigation:** Yasmine Khan.

**Methodology:** Yasmine Khan, Nick Verhaeghe, Delphine De Smedt.

**Resources:** Yasmine Khan.

**Supervision:** Nick Verhaeghe, Brecht Devleesschauwer, Sylvie Gadeyne, Delphine De Smedt.

**Validation:** Yasmine Khan, Nick Verhaeghe, Delphine De Smedt.

**Writing – original draft:** Yasmine Khan.

**Writing – review & editing:** Yasmine Khan, Nick Verhaeghe, Robby De Pauw, Brecht Devleesschauwer, Sylvie Gadeyne, Vanessa Gorasso, Yolande Lievens, Niko Speybroek, Nancy Vandamme, Miet Vandemaele, Laura Van den Borre, Sophie Vandepitte, Katrien Vanthomme, Freija Verdoodt, Delphine De Smedt.

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
