## [Decision Letter · Decision Letter 0]

27 Mar 2023

PONE-D-22-31778Evaluating the health and health economic impact of the COVID-19 pandemic on delayed cancer care in Belgium: A Markov model study protocolPLOS ONE

Dear Dr. Khan, Thank you for submitting your manuscript to PLOS ONE. After careful consideration, we feel that it has merit but does not fully meet PLOS ONE’s publication criteria as it currently stands. Therefore, we invite you to submit a revised version of the manuscript that addresses the points raised during the review process. Please note that I have acted as a reviewer for this manuscript, and you will find my comments below, under Reviewer #2.

We look forward to receiving your revised manuscript.

Kind regards,

Matteo Ruggeri

Academic Editor

PLOS ONE

Journal Requirements:

2. Our staff editors have determined that your manuscript is likely within the scope of our Early Detection, Screening and Diagnosis of Cancer Call for Papers. This editorial initiative is headed by in-house PLOS editors. This Call for Papers aims to explore recent advances in the early detection of cancer and implications of these advances for patient survival. Additional information can be found on our announcement page: https://collections.plos.org/call-for-papers/early-detection-screening-and-diagnosis-of-cancer/

If you would like your manuscript to be considered for this collection, please let us know in your cover letter and we will ensure that your paper is treated as if you were responding to this call. Please note that being considered for the Call for Papers does not require additional peer review beyond the journal’s standard process and will not delay the publication of your manuscript if it is accepted by PLOS ONE. If you would prefer to remove your manuscript from collection consideration, please specify this in the cover letter

3. Please confirm that all data sources you used were publicly available and anonymized. If this is not the case, please provide information on what permissions you were granted to access these data.

Reviewers' comments:

Reviewer's Responses to Questions

**Comments to the Author**

1. Does the manuscript provide a valid rationale for the proposed study, with clearly identified and justified research questions?

Reviewer #1: Partly

Reviewer #2: Yes

2. Is the protocol technically sound and planned in a manner that will lead to a meaningful outcome and allow testing the stated hypotheses?

Reviewer #1: Partly

Reviewer #2: Yes

3. Is the methodology feasible and described in sufficient detail to allow the work to be replicable?

Reviewer #1: No

Reviewer #2: Yes

4. Have the authors described where all data underlying the findings will be made available when the study is complete?

Reviewer #1: No

Reviewer #2: Yes

5. Is the manuscript presented in an intelligible fashion and written in standard English?

Reviewer #1: Yes

Reviewer #2: Yes

6. Review Comments to the Author

You may also provide optional suggestions and comments to authors that they might find helpful in planning their study.

Reviewer #1: The paper “Evaluating the health and health economic impact of the COVID-19 pandemic on delayed cancer care in Belgium: A Markov model study protocol” presents a protocol for investigating the health and economic impact of delaying cancer diagnosis due to the pandemic. The protocol deals with all the relevant aspect of such a kind of analysis, even though it mostly relies on a teaching approach that could be considered rude for expert readers. In fact, the main concern of the protocol is that it only presents the underlying methodology. I would have expected to see some input parameters already identified and reported and more information provided on crucial aspects of the analysis. In respect to the latter aspect, Authors mentioned a matched case-control study that is not thoroughly described in its aim and methodology. I guess that querying the BRC would serve to understand how distribution among states has been affected by covid, but this is not clear. Furthermore, Authors spoke about two Belgian experts, but they did not provide any information about their background and selection. Eventually, introduction could be enriched of current literature on the delays (and their impacts) due to the covid also to better disentangle the rationale behind the study and the research gap that it is going to meet.

Reviewer #2: The study protocol is well written and referenced. Methods are technically sound and robust. However, the protocol follows a very traditional approach and I do not see actually a valid reason for publication. 

For a protocol to be published this should include an innovative methodology or a soecial interest for other readers to reply the protocol and therefore compare results, which is not the case at this time.

Authors should publish the manuscript ad full research article once results are ready.

7. PLOS authors have the option to publish the peer review history of their article (what does this mean?). If published, this will include your full peer review and any attached files.

Reviewer #1: No

Reviewer #2: No

---

## [Author Response · Author response to Decision Letter 0]

26 May 2023

Manuscript title: 

Evaluating the health and health economic impact of the COVID-19 pandemic on delayed cancer care in Belgium: A Markov model study protocol

Authors:

Yasmine Khan, Nick Verhaeghe, Robby De Pauw, Brecht Devleesschauwer, Sylvie Gadeyne, Vanessa Gorasso, Yolande Lievens, Niko Speybroek, Nancy Vandamme, Miet Vandemaele, Laura Van den Borre, Sophie Vandepitte, Katrien Vanthomme, Freija Verdoodt, Delphine De Smedt

Dear Editor,

Dear Reviewers,

We appreciate the valuable comments and suggestions you provided for our manuscript, which we have considered carefully. Please refer to our response below, where we address each point raised (indicated in bold and italic). Additionally, we have incorporated the changes in the manuscript, which are highlighted (in italic and underlined) in our response. Furthermore, we updated the manuscript because of an important development regarding data availability. The Belgian Cancer Registry has notified the authors that the data regarding the skin melanoma cancer type cannot be delivered due to quality control issues. Hence, we have suppressed all information regarding skin melanoma cancer from the manuscript. 

Reviewer #1: The paper “Evaluating the health and health economic impact of the COVID-19 pandemic on delayed cancer care in Belgium: A Markov model study protocol” presents a protocol for investigating the health and economic impact of delaying cancer diagnosis due to the pandemic. The protocol deals with all the relevant aspect of such a kind of analysis, even though it mostly relies on a teaching approach that could be considered rude for expert readers. In fact, the main concern of the protocol is that it only presents the underlying methodology. I would have expected to see some input parameters already identified and reported and more information provided on crucial aspects of the analysis. In respect to the latter aspect, Authors mentioned a matched case-control study that is not thoroughly described in its aim and methodology. I guess that querying the BRC would serve to understand how distribution among states has been affected by covid, but this is not clear. Furthermore, Authors spoke about two Belgian experts, but they did not provide any information about their background and selection. Eventually, introduction could be enriched of current literature on the delays (and their impacts) due to the covid also to better disentangle the rationale behind the study and the research gap that it is going to meet.

Detail of comment: 

“I would have expected to see some input parameters already identified and reported and more information provided on crucial aspects of the analysis. In respect to the latter aspect, Authors mentioned a matched case-control study that is not thoroughly described in its aim and methodology. I guess that querying the BRC would serve to understand how distribution among states has been affected by covid, but this is not clear.”

Reply:

Thank you for your comment. At the moment, input parameters cannot be identified and described accurately by the Authors. When the data will be delivered, we will review the data quality of the requested variables. We will then provide comprehensive details on the input parameters and the data sources, together with the results. Our goal for this manuscript was to provide a methodological framework that enables researchers to evaluate the impact of the COVID-19 pandemic on delayed cancer care from a health-economic perspective in their own country. We believe that presenting the methodology before conducting the analyses is worthwhile to provide a clear, transparent and replicable framework. 

We have added more information on the matched case-control study from line 236-250 as follows:

In a parallel study, cancer costs are estimated at the individual level for each considered cancer site and each cancer stage with the method of recycled predictions (also known as the method of standardisation or g-computation), which determines the marginal difference in costs for a sample of patients being affected with one type of cancer compared to a sample of the general population who do not present any cancer diagnosis (39–41). The aim is to estimate the average total healthcare expenditure during one year, being 2018 (the most recent year for which data is available) for cancer patients according to their cancer site and their cancer stage, which will be used as input in the model. The cancer population consists of a sample of the Belgian cancer population alive on the 1st January 2018, and who had at least one cancer diagnosis of the considered cancers during the 10 previous years (S1 Appendix). The control sample is selected from the healthcare insurer database (IMA) if they were alive on the 1st January 2018 and were not diagnosed with cancer in the 10 years preceding 2018. In order to limit the amount of individuals, a random subset of the patient population is made. Matching is done based on age, sex, and region of residence available on the 1st January 2018. A case-control ratio of 1:4 is used, considering that including more than four controls per case does not increase significantly the statistical power (42).

Detail of comment: 

 “Furthermore, Authors spoke about two Belgian experts, but they did not provide any information about their background and selection.”

Reply:

We agree with this comment and have specified (line 119-127): 

To know whether the model is a good reflection of reality, validation is sought by two Belgian experts. Experts were selected using following criteria: at least 10 years of medical practice and active research experience in the field of cancer health economics in Belgium. Expert A has been working as an oncologist for over 20 years and has received health economic training. His research expertise includes the impact of the COVID-19 pandemic on delayed cancer diagnosis and treatment in Belgium. Expert B is an expert in radiation oncology and in health economics. Her work experience includes the organizational aspects of cancer treatment, including the financial and health economic aspects of cancer care. 

Detail of comment: 

“Eventually, introduction could be enriched of current literature on the delays (and their impacts)

due to the covid also to better disentangle the rationale behind the study and the research gap that it is going to meet.”

Reply:

We appreciate your suggestion. We have subsequently revised the introduction and included new and recent references. We have incorporated this in line 85-94 as follows:

A recent study conducted by the Madrid tumor registry analysed cancer diagnoses between 2019-2021 in Spain. The study evaluated the differences in annual volume, observed-to-expected (O/E) volume ratios, and standardised incidence rate ratios for 2020-2021 compared to 2019, which was found to be 94.5% (95% CI 93.8-95.3). The study revealed that most cancer types were underdiagnosed in 2020, and this trend worsened in 2021 for colorectal and prostate cancers (87.8%). However, lung cancer showed improvement (102.1%), and breast cancer was overdiagnosed (114.4%) compared to pre-COVID-19 reference data (20). Another study conducted in Japan found more aggressive and advanced disease after the suspension of breast cancer screening services during the COVID-19 pandemic with the percentage of stage IIB or higher patients being significantly higher in the pandemic group than in the non-pandemic group (22.0% vs 31.3%, p= 0.0133) (21).

Reviewer #2: The study protocol is well written and referenced. Methods are technically sound and robust. However, the protocol follows a very traditional approach and I do not see actually a valid reason for publication. 

For a protocol to be published this should include an innovative methodology or a soecial interest for other readers to reply the protocol and therefore compare results, which is not the case at this time.

Authors should publish the manuscript ad full research article once results are ready.

Reply:

Thank you for taking the time to review our study protocol and for your positive feedback. We appreciate your comment regarding the traditional approach of our protocol, but we would like to clarify that our objective was not to introduce a new or innovative methodology. Instead, our aim was to provide a reproducible and accessible framework that could be used by researchers to evaluate the impact of the COVID-19 pandemic on delayed cancer care from a health-economic perspective in their own country., instead of waiting for long-time observational data. 

Our study protocol fills a gap in the literature by providing a detailed and comprehensive description of the methodology and analytical approach we intend to use in our subsequent research article and that may also serve as inspiration for such future research. We conducted a thorough literature review and had discussions with high-value oncologists to ensure that our protocol follows the best practices in the field. Our goal was to develop a framework that other researchers could utilise to evaluate the health-economic impact of the pandemic on cancer care. We recognise the significance of transparency in research and guarantee to provide a comprehensive explanation of our methods.

We acknowledge that our protocol does not provide results or input parameters from a specific study, but we believe that it is an essential first step in evaluating the impact of the pandemic on cancer care. Without a comprehensive study protocol, it may be challenging for researchers to evaluate the methods and analytical approach used in our study. The 2022 version of the CHEERS checklist now includes an item called “Health economic analysis plan” to indicate whether a health economic analysis plan was developed and where to find it, which highlights the importance of developing a robust protocol before diving into health economic evaluations (Husereau et al., 2022). 

References:

Husereau, D., Drummond, M., Augustovski, F., de Bekker-Grob, E., Briggs, A. H., Carswell, C., Caulley, L., Chaiyakunapruk, N., Greenberg, D., Loder, E., Mauskopf, J., Mullins, C. D., Petrou, S., Pwu, R.-F., Staniszewska, S., & on behalf of CHEERS 2022 ISPOR Good Research Practices Task Force. (2022). Consolidated Health Economic Evaluation Reporting Standards 2022 (CHEERS 2022) statement : Updated reporting guidance for health economic evaluations. BMC Medicine, 20(1), 23. https://doi.org/10.1186/s12916-021-02204-0

---

## [Editor Report · Decision Letter 1]

4 Jul 2023

Evaluating the health and health economic impact of the COVID-19 pandemic on delayed cancer care in Belgium: A Markov model study protocol

PONE-D-22-31778R1

Dear Dr. Kahn

We’re pleased to inform you that your manuscript has been judged scientifically suitable for publication and will be formally accepted for publication once it meets all outstanding technical requirements.

Kind regards,

Matteo Ruggeri

Academic Editor

PLOS ONE

---

## [Editor Report · Acceptance letter]

10 Jul 2023

PONE-D-22-31778R1 

Evaluating the health and health economic impact of the COVID-19 pandemic on delayed cancer care in Belgium: A Markov model study protocol 

Dear Dr. Khan:

I'm pleased to inform you that your manuscript has been deemed suitable for publication in PLOS ONE. Congratulations! Your manuscript is now with our production department. 

Kind regards, 

on behalf of

Dr. Matteo Ruggeri 

Academic Editor

PLOS ONE